# Obesity Status and Physical Fitness Levels in Male and Female Portuguese Adolescents: A Two-Way Multivariate Analysis

**DOI:** 10.3390/ijerph20126115

**Published:** 2023-06-13

**Authors:** Samuel Encarnação, Filipe Rodrigues, António Miguel Monteiro, Hatem Gouili, Soukaina Hattabi, Andrew Sortwell, Luís Branquinho, José Eduardo Teixeira, Ricardo Ferraz, Pedro Flores, Sandra Silva-Santos, Joana Ribeiro, Amanda Batista, Pedro Miguel Forte

**Affiliations:** 1Department of Sports Sciences, Universidad Autónoma de Madrid (UAM), 28049 Madrid, Spain; 2Department of Sport Sciences, Instituto Politécnico de Bragança (IPB), 5300-253 Bragança, Portugal; 3Research Centre in Sports Sciences, Health Sciences and Human Development (CIDESD), 5001-801 Vila Real, Portugal; 4Research Center of the Higher Institute of Educational Sciences of the Douro, (CI-ISCE), 4560-447 Penafiel, Portugal; 5ESECS—Polytechnic of Leiria, 2411-901 Leiria, Portugal; 6Life Quality Research Center (CIEQV), 2040-413 Leiria, Portugal; 7High Institute of Sports and Physical Education of Elkef, University of Jendouba, Jendouba Kef 7100, Tunisia; 8Higher Institute of Educational Sciences of the Douro, 4560-447 Penafiel, Portugal; 9School of Nursing, Midwifery, Health Sciences and Physiotherapy, University of Notre Dame, Sydney 2007, Australia; 10Department of Sport Sciences, Polytechnic Institute of Guarda, 6300-559 Guarda, Portugal; 11Department of Sports Sciences, University of Beria Interior, 6201-001 Covilhã, Portugal; 12Research Center in Sports Performance, Recreation, Innovation and Technology (SPRINT -IPVC), 4900-347 Viana do Castelo, Portugal; 13Instituto Politécnico de Viana do Castelo (IPVC), 4900-347 Viana do Castelo, Portugal

**Keywords:** adolescence, metabolic syndrome, non-communicable chronic disease, immune system, inflammation, quality of life

## Abstract

Obesity and decreasing fitness levels among the youth are growing concerns in Portugal, similar to other developed countries, with implications for health and psychomotor development. Understanding the influence of health determinants such as sex and age are crucial for developing effective public health strategies. This study aimed to analyze the association between sex and chronological age with obesity status and physical fitness in Portuguese adolescents. A total of 170 adolescents (85 males and 85 females) were evaluated for body mass index, abdominal adiposity, aerobic fitness, abdominal resistance, upper limb resistance, lower limb power, and maximal running speed in a 40 m sprint using the FITescola^®^ physical fitness battery, a Portuguese government initiative. The general model, analyzed using Pillai’s trace, showed a significant effect of age and sex on body mass index, abdominal circumference, aerobic fitness, abdominal resistance, upper limb resistance, lower limb power, and maximal running speed (V = 0.99, F (7) = 10,916.4, *p* < 0.001, partial η2, sex = 0.22; age = 0.43, sex and age interaction = 0.10). Boys had higher physical fitness levels than girls in most tests, but both sex groups had a significantly higher proportion of non-fit adolescents, with boys showing the highest number of participants classified as non-fit.

## 1. Introduction

Childhood and adolescent obesity pose significant risks for serious health complications [1]. These risks are associated with an elevated and chronic inflammatory profile in the body, leading to homeostatic inflammation, which involves altered immune responses triggered by cell damage resulting from increased inflammation [2]. These immune alterations negatively affect metabolism, leading to metabolic syndrome and its associated complications, such as diabetes and cardiovascular disease [3,4]. Moreover, metabolic deregulation increases the risk of cancer development and premature death [5]. On the other hand, physical fitness is a crucial determinant of health and well-being and can be developed and improved from infancy throughout the lifespan [6]. Physical fitness has been shown to protect against physical inactivity and sedentary lifestyles, which are risk factors for non-communicable diseases, and it can also slow the progression of chronic diseases [7].

Population-based studies conducted in several countries have consistently demonstrated a robust association between poor physical fitness during childhood, adulthood, and later stages of life [8]. Inadequate physical fitness has been shown to contribute to a 23% significantly increased risk of various types of cancer and a 57% heightened risk of cardiovascular disease, persisting into adulthood. These risks are particularly prevalent among individuals who do not meet the minimal physical activity levels recommended by the World Health Organization (WHO) and exceed the recommended limits of sedentary behavior time [7,8,9].

Two broad categories of factors influence adolescent obesity risk: modifiable and non-modifiable risk factors [9]. Non-modifiable risk factors encompass genetic predisposition, family history, race, and ethnicity, which are beyond an individual’s control [9]. On the other hand, modifiable factors include physical activity levels, previous sports participation, sedentary behavior, duration of electronic device use, and dietary behavior, which are considered pivotal factors associated with the development of excessive weight and obesity [9]. 

Furthermore, lifestyles can vary significantly based on geographical location, influenced by cultural and socioeconomic factors [6]. A country’s culture can undergo significant changes depending on its income level, with higher educational levels commonly observed in high-income countries [10]. Data from over 100 countries with varying income levels have shown that socioeconomic factors may profoundly impact the prevalence of obesity. In some instances, individuals in high-income countries may have access to a diverse range of physical activities compared to those in low-income localities. However, due to the prevalence of easily accessible fast-food options in high-income countries, obesity rates may also be higher compared to low-income countries. Thus, the evidence suggests that besides income levels, cultural and behavioral factors also significantly influence the risk of obesity [10].

Obesity levels exhibit a higher prevalence in females, and recent research in the general population suggests that this sex disparity may vary across different countries, influenced by socioeconomic and cultural factors [11]. Furthermore, epidemiological data have shown that sex differences in obesity are more pronounced in low-income countries [11]. In Portugal, the research on modifiable and non-modifiable factors in the incidence and prevalence of obesity and sex influences is limited [12,13]. Therefore, conducting studies that analyze the influence of sex could provide valuable insights to enhance the quality and quantity of information on this topic, thereby stimulating new perspectives and comprehensive implications for research on obesity and its modifiable and non-modifiable factors in Portuguese adolescents. 

This study aimed to investigate the obesity status and physical fitness levels in male and female adolescents using a multivariate analysis. Our main research hypotheses were as follows: (a) girls would exhibit higher rates of obesity compared to boys; (b) among girls, those in middle adolescence would display higher rates of obesity compared to younger girls, and boys from all age groups would exhibit similar patterns; and (c) girls would demonstrate poorer physical fitness levels compared to boys.

## 2. Materials and Methods

### 2.1. Study Participants

The present study utilized R, a commonly employed programming language for statistical computing, to determine the appropriate sample size for a MANOVA design with repeated measures between factors [14]. Input parameters were specified, including an expected effect size of 0.25, an alpha level of 0.05, a power of 0.95, and a correlation among repeated measures of 0.50. The study aimed to include two groups with two measurements in total. The analysis revealed that a minimum of 106 participants were required to achieve adequate statistical power.

In this cross-sectional investigation, we aimed to examine the relationship between sex, age, obesity, and physical fitness in Portuguese adolescents. The study enrolled a total of 170 adolescents aged 10–16 years (mean age of 13 ± 2.40 years), with an average height of 1.56 ± 11.00 cm and a body weight of 50.70 ± 13.70 kg, Table 1. The eligibility criteria included adolescents of both sex, without any disabling conditions, and within the age range of 10–16 years, in line with the WHO cut-off criteria.

### 2.2. Ethical Aspects

The study in question was granted approval by the Scientific Board of the Higher Institute of Educational Sciences of the Douro (PF: 10.2021). Before data collection, the study’s goals were comprehensively explained to every parent or legal guardian, and written informed consent was acquired from each person. Moreover, all minor participants were also requested to provide written informed consent after obtaining approval from their parents or legal guardians.

### 2.3. Data Collection

In this study, an extensive set of measures and tests were used to evaluate obesity status and physical fitness. To assess obesity status, body mass index (BMI) and abdominal adiposity were utilized. Physical fitness data were collected from the FITescola^®^ tests [15,16], a Portuguese project developed to assess and promote healthy behaviors in children and adolescents, administered in September 2021. Multiple tests were employed to evaluate physical fitness, including the Yo-Yo test to measure aerobic fitness; abdominal curl to evaluate abdominal resistance; push-up test to assess upper limb resistance; lower limb power; and the 40 m sprint time test to determine maximal running speed. The FITescola^®^ battery tests have been previously validated in the context of physical education and sport [17].

### 2.4. Measurements

To obtain body weight measurements, participants were weighed while barefoot and wearing light clothing, standing upright, and waiting for the score on the brand scale to stabilize. A brand scale with a precision of 100 g was used for this purpose. Height measurements were taken with participants standing barefoot, with their feet together, and their back touching the stadiometer scale. The stadiometer hod was positioned at the top of the participant’s head to compress the highest part of their head. A stadiometer with a precision of millimeters was used for height measurements. BMI was calculated by dividing body weight by height squared (kg/m^2^). The cut-off values for the risk of cardiovascular disease were established following the norms set by the WHO for adolescents [18].

Abdominal adiposity was assessed by measuring waist circumference. Participants were instructed to stand upright with a relaxed belly and lift their shirt to expose the area to be measured. A measuring tape was then placed around the waist, in the horizontal plane, 1 cm above the top of the iliac crest. Participants were asked to perform a normal expiration, and the value obtained at the end of expiration was recorded in centimeters (cm) using a precision tape measure with a measurement accuracy of 0.1 cm. Two attempts were made, and the average of the two measurements was considered the final evaluation result [17]. The cut-off values for cardiovascular disease risk were determined based on the norms set by the International Diabetes Federation criteria for abdominal circumference in adolescents [19].

The study utilized established protocol to administer the Yo-Yo test, which is a widely used assessment tool for measuring aerobic fitness [17]. The test involved positioning two cones 20 m apart and the participant standing behind the starting line. The participant began running upon hearing the audio-generated sound signal and was required to touch the 20-meter line before reversing in direction and running back to the starting line upon hearing the subsequent sound signal. The audio signal served as a reference for monitoring the speed during the test, and the initial running speed was set at 8.5 km/h. The running speed was increased progressively by 0.5 km/h every minute until reaching a maximum of 120 rounds. The test result was determined by the highest number of laps performed, and the first failed attempt was recorded as the final score [17].

In this study, the abdominal curl test was used to measure abdominal resistance according to established procedures [17]. The participant began the test in a supine position on the ground, with knees flexed at an approximate angle of 140°, and feet positioned approximately hip-width apart. The participant then performed trunk flexion, controlling both the concentric and eccentric phases of the movement until their hands touched their knees. The test was considered complete when the participant could no longer perform repetitions or when the maximal cut-off of 75 repetitions was reached.

In this study, the push-up test was conducted in accordance with established protocols [17] to evaluate upper limb resistance. The participant was instructed to assume a plank position with their feet positioned hip-width apart and tiptoes touching the ground. Their hands were placed directly on the shoulder line with fingers pointing forward, in the starting position. During the test, the participant was required to maintain the plank position of the trunk. The movement was initiated by flexing the elbows at a pace of 1 s for the eccentric phase, lowering the body toward the ground, and then extending the elbows fully at a pace of 1 s, returning to the starting position. This sequence was repeated until the participant reached muscle failure, and the evaluator recorded the maximal number of repetitions achieved.

To assess lower limb power, a horizontal line was drawn at the starting point and reference lines were drawn every 10 cm (1 m after the starting line). A measuring tape with an accuracy of 1 mm was placed perpendicular to the horizontal lines to facilitate the measurement of the distance reached. The participant stood behind the line marking the starting point with their feet positioned shoulder-width apart. Starting from a standing position, the participant bent their knees, pulled their arms behind their back, and jumped as far as possible. The distance was measured from the restarting point to the heel, and two attempts were made to record the best result of the two evaluations in cm [17].

The evaluation of maximal running speed over a distance of 40 m was conducted using a standardized 40-meter sprint test [17]. Prior to the test, a 3-minute warm-up was administered to ensure general muscle activation and reduce the risk of injuries. Two signaling cones were placed to demarcate the initial and final test courses. The participant assumed a standing position behind the starting line, with their lower limbs positioned in anteroposterior alignment and their trunk slightly inclined forward. At the evaluator’s “prepare, now” signal, the participant initiated the sprint at the highest possible speed. The stopwatch was stopped when the participant crossed the finish line. Two trials were conducted, and the best-recorded result was documented for further analysis [17].

### 2.5. Statistical Analysis

R, a programming language designed for statistical computing, was utilized for the statistical analyses in this study [14]. Descriptive statistics, including means and standard deviations, were used to summarize the characteristics of the study sample and continuous data. Proportions of adolescents across different groups were reported as absolute and percentage values. Differences in the proportions of obese adolescents between groups were assessed using a chi-squared test (χ^2^) for two proportions. To investigate the relationship between age, sex, obesity status, and physical fitness, a two-way between-subjects multivariate analysis of variance (MANOVA) was performed. Before conducting the MANOVA, the linearity between independent variables was assessed to detect potential multicollinearity. The presence of multivariate outliers was evaluated using the Mahalanobis distance method [20]. To confirm the presence of true outliers in each dependent variable in accordance with existing assumptions, a readjusted significance level of *p* > 0.01 for the χ^2^ was used [20]. Outliers were removed from the dataset if identified to enhance the precision of the analysis. The homogeneity of covariance matrices was assessed using Box’s M test, assuming equal covariances across groups [15]. Multivariate normality was assessed using the Shapiro–Wilk normality test, adjusted for two-way MANOVA [16]. Finally, a two-way MANOVA was conducted with “age” and “sex” as dependent variables, and “obesity status” and “physical fitness” as independent variables for adolescents. The Pillai’s trace was used to identify significant paired comparisons between dependent and independent variables [21]. Effect sizes were calculated using partial eta squared (ηp^2^) with small = 0.10, moderate = 0.30, and large = 0.50 cut-offs, as recommended by Cohen [22]. 

## 3. Results

In the 10-year-old age group, a statistically significant higher number of participants were classified as obese based on BMI cut-offs (normal weight, n = 27 (46.4%); obesity, n = 31 (58.6%), X^2^ = 19.7, *p* < 0.001, small effect size = 0.22). Despite statistically significant numbers of obese participants in the 13-year-old and 16-year-old groups (*p* < 0.05), the rates of obesity remained high (13 years, n = 26 (43%); 16 years, n = 15 (29%)). The general model showed a significant effect of age and sex on the independent variables of BMI, abdominal circumference, aerobic fitness, abdominal resistance, upper limb resistance, lower limb power, and the 40 m sprint (Pillais trace = 0.99, F (7) = 10,916.4, *p* < 0.001, effect size of sex = 0.22; effect size of age = 0.43; effect size of sex and age interaction = 0.10).

For obesity status, it was observed that only age had a significant and positive effect on BMI (F (2) = 15.2, *p* > 0.001), where adolescents aged 16 and 13 years had higher means of BMI compared to the 10 years group (10 years: 18 ± 2.95; 13 years: 21 ± 3.63; 16 years: 22 ± years, see Figure 1A). There were no interactions between age and sex (F (2) = 1.45, *p* = 0.23). The proportion of girls with obesity in the 13 years group was significantly higher than in boys (girls 13 years: 26%, n = 8; boys 13 years: 6%, n = 2, X^2^ = 85.767, df = 1, *p* < 0.0001, large effect size, V = 0.92).

Regarding abdominal adiposity, there was a significant effect of age on BMI (F (2) = 13.94, *p* < 0.001), with adolescents aged 13 and 16 years showing higher means of abdominal adiposity compared to the 10 years group. There were no interactions between age and sex (F (2) = 1.86, *p* = 0.15); see Figure 1B. There was a statistically significant higher proportion of girls with abdominal obesity compared to boys in all three age groups (girls 10 years: 37%, n = 11; boys 10 years: 7%, n = 2, X^2^ = 7.413, df = 1, *p* < 0.001, moderate effect size, V = 0.28; girls 13 years: 23%, n = 7; boys 13 years: 10%, n = 3, X^2^ = 15.959, df = 1, *p* < 0.001, moderate effect size, V = 0.40; girls 16 years: 42%, n = 11; boys 16 years: 4%, n = 1, X^2^ = 15.959, df = 1, *p* < 0.001, large effect size, V = 0.71).

In terms of aerobic fitness, the results revealed a statistically significant effect of age (F (2) = 60.99, *p* < 0.001), with higher average cycles completed in the Yo-Yo test observed in the 13-year-old and 16-year-old age groups. Furthermore, a significant sex effect was also observed (F (1) = 16.38, *p* < 0.001), with males exhibiting a higher average performance compared to females. Additionally, a statistically significant interaction between age and sex was identified (F (2) = 4.76, *p* < 0.05), whereby boys in all three age groups displayed higher average scores. Notably, besides the interaction between age and sex, a significantly higher proportion of girls compared to boys were classified as non-fit in the 10-year-old and 13-year-old age groups (girls 10 years: 51%, n = 15, boys 10 years: 34%, n = 10, X^2^ = 32.01, df = 1, *p* < 0.001, moderate effect size, V = 0.30; girls 13 years: 33%, n = 10, boys 13 years: 13%, n = 4, X^2^ = 34.3, df = 1, *p* < 0.001, moderate effect size, V = 0.30), as illustrated in Figure 2.

In terms of abdominal resistance, the findings revealed a statistically significant isolated effect of sex (F (2) = 8.30, *p* < 0.001), with boys exhibiting a higher average performance compared to girls. Additionally, a significant effect of age was observed (F (2) = 15.19, *p* < 0.001), with higher average scores observed in the 13-year-old and 16-year-old age groups of both sex compared to the 10 years group. Furthermore, a statistically significant interaction between sex and age was identified (F (2) = 3.22, *p* < 0.05), whereby boys in the 13-year-old and 16-year-old age groups displayed higher average scores compared to girls, while girls in the 10-year-old age group showed higher average scores compared to boys. Additionally, there was a significant but small effect size difference in the proportion of fit and non-fit boys compared to girls in the 10-year-old age group (girls 10 yo: 24%, n = 7, boys 10 yo: 27%, n = 8, X^2^ = 7.072, df = 1, *p*-value = 0.01, small effect size, V = 0.10). No differences were observed in the 13-year-old age group (girls 13 years: 30%, n = 9, boys 13 years: 30%, n = 9, X2 = 0, df = 1, *p* > 0.05, no effect size, V = 0), while a statistically significant proportion of non-fit boys compared to girls was observed in the 16-year-old age group (girls 16 years: 11%, n = 3, boys 16 years: 23%, n = 6, X^2^ = 52.04, df = 1, *p* < 0.001, moderate effect size, V = 0.40), as illustrated in Figure 3.

No statistically significant isolated effects between sex or age groups were found in relation to upper limb resistance, with *p*-values exceeding 0.05. However, a statistically significant interaction effect between sex and age was observed (F (2) = 5.09, *p* < 0.01), wherein boys in the 16-year-old age group demonstrated higher average scores in push-ups compared to girls. Furthermore, a statistically significantly higher proportion of non-fit boys than girls was observed across all age groups. Specifically, in the 10-year-old age group, boys had a non-fit rate of 31% (n = 9) while girls had a rate of 27.5% (n = 8) (X^2^ = 9.3076, df = 1, *p* < 0.001, small effect size, V = 0.15). In the 13-year-old age group, boys had a 100% non-fit rate (n = 30), while girls had a rate of 46.6% (n = 14) (X^2^ = 71.258, df = 1, *p* < 0.001, moderate effect size, V = 0.42). Lastly, in the 16-year-old age group, boys had a 100% non-fit rate (n = 26), while girls had a rate of 27% (n = 7) (X^2^ = 111.83, df = 1, *p* < 0.001, large effect size, V = 0.58), as illustrated in Figure 4.

Significant isolated effects of sex (F (1) = 28.7, *p* < 0.001) were observed in relation to lower limb power, with boys displaying higher average distances than girls across all age groups. Age also had a significant isolated effect (F (2) = 37.9, *p* < 0.001), wherein 13-year-olds and 16-year-olds demonstrated higher averages compared to the 10-year-olds (10 years: 124 ± 6 cm; 13 yo: 152 ± 16 cm; 16 years: 156 ± 14 repetitions). However, no statistically significant interaction between sex and age was found (*p* > 0.05). Nevertheless, a notable proportion of non-fit girls compared to boys was observed in the 10-year-old age group, with 24% of girls (n = 7) being non-fit compared to 10% of boys (n = 3) (X^2^ = 60.06, df = 1, *p* < 0.001, moderate effect size, V = 0.43). Conversely, a higher proportion of non-fit boys compared to girls was observed in the 16-year-old age group, with 77% of boys (n = 20) being non-fit compared to 8% of girls (n = 2) (X^2^ = 143.12, df = 1, *p* < 0.001, large effect size, V = 0.66). No differences in the proportion of non-fit adolescents were found in the 13-year-old age group, with both boys and girls having a non-fit rate of 17% (n = 5) (X^2^ = 0, df = 1, *p* > 0.05, no effect size, V = 0), as illustrated in Figure 5.

Regarding the maximum running speed of adolescents, there was a statistically significant isolated effect of sex (F (1) = 20.64, *p* < 0.001), wherein boys aged 13 and 16 years displayed shorter sprint times compared to girls. Additionally, there was a significant isolated effect of age (F (2) = 52.42, *p* < 0.001), with both boys and girls aged 13 and 16 years demonstrating shorter sprint times than the 10-year-old age group. Furthermore, no statistically significant interactions were found between sex and age (*p* > 0.05). There was a higher proportion of non-fit boys compared to girls in the 10-year-old and 16-year-old age groups, with 41% of boys (n = 12) and 61% of boys (n = 16) being non-fit, respectively, compared to 34% of girls (n = 10) and 33% of girls (n = 3) (X^2^ = 14.529, df = 1, *p*-value < 0.001, small effect size, V = 0.13; X^2^ = 52.41, df = 1, *p* > 0.0001, moderate effect size, V = 0.41), as illustrated in Figure 6. On the contrary, a significantly higher proportion of non-fit girls compared to boys was found in the 13-year-old age group, with 40% of girls (n = 10) and 36% of boys (n = 11) being non-fit (X^2^ = 7.006, df = 1, *p* < 0.01, small effect size, V = 0.13).

## 4. Discussion

This research aimed to investigate the prevalence of obesity and physical fitness in adolescents from Portugal. The study hypotheses (a) and (b) were found to be incorrect as adolescents aged 13–16 had higher BMI scores and abdominal adiposity regardless of sex, and both sex had high obesity rates (44% in boys and 41% in girls). The third hypothesis (c) was supported, as the 13–16 age groups of both sex showed better aerobic fitness, with boys demonstrating greater aerobic fitness in all age groups. Boys also had greater abdominal resistance and lower limb power than girls in all age groups, and greater upper limb resistance than girls in the 16-year-old group. Additionally, boys had better sprint performances than girls in all the age groups. However, both sex had higher proportions of non-fit adolescents, with boys showing the highest incidence.

The findings of this study indicate that age has a significant positive effect on BMI, as shown by the statistical analysis (F (2) = 15.2, *p* > 0.001). Specifically, adolescents aged 13 and 16 years had higher BMI scores than 10-year-olds. Furthermore, a greater proportion of girls in the 13-year-old group were obese compared to boys, and this sex difference was reflected in a moderated effect size. It is well known that hormonal changes during adolescence tend to lead to greater adipose tissue accumulation in girls than in boys [23]. Nevertheless, the obesity classification used in this study is considered to be a sex-neutral indication of a pathogenic health state [24]. In addition, the study emphasizes the importance of modifiable behavior factors such as promoting physical activity, healthy nutrition, and lifestyle choices in influencing the childhood obesity phenotype [25]. This is consistent with the findings of a recent systematic review by Narciso et al. [6], which analyzed 40 prospective cohort studies conducted between 2000 and 2018 and concluded that socioeconomic status is a significant predictor of adolescent obesity.

The results of this study suggest that a higher BMI is associated with metabolic deregulation, an increased inflammatory profile, a reduced immune system, and insulin resistance. These factors are key in the onset of type II diabetes during childhood and in the development of cardiovascular diseases in adulthood [19,26]. Moreover, a high BMI can impair proper bone and muscle system development during the growth and maturation process [3,25]. The manifestation of a high BMI in late adolescence implies a negative transition from a healthy state to adulthood, which poses a risk to multiple systems [4]. Additionally, the evidence suggests that obese adolescents are more likely to be bullied in the school environment, putting their quality of life and self-esteem at risk [8]. Without early preventive interventions, these individuals may suffer serious mental sequelae that could last for the rest of their lives [8].

The study also found that abdominal adiposity had a significant isolated effect of age, where adolescents aged 13 and 16 years had higher abdominal adiposity than those aged 10 years. Furthermore, even without significant interactions between age and sex, there was a statistically significant higher proportion of girls with abdominal obesity than boys in all the three age groups, as reflected by a moderate effect size. In contrast, a cross-sectional study conducted in Wales by Lewitt and Baker [27] found that boys had higher abdominal adiposity and inflammatory profiles than girls, despite having a similar BMI. Interestingly, the maternal and paternal BMI also influenced abdominal obesity status. Elevated abdominal adiposity is of particular concern as it implies deposits of adipose cells in visceral tissues, which is directly associated with insulin resistance and early metabolic syndrome [27]. This condition is associated with childhood type II diabetes and heart disease [19].

Furthermore, elevated abdominal adiposity has been linked to increased systemic inflammation, which creates a pro-tumoral environment and is associated with childhood cancer [28]. In terms of mental health, high levels of abdominal adiposity have been linked to poor sleep quality [29], which is associated with the development of mental disorders such as major depressive disorders at an early age [19]. Additionally, the obese phenotype can negatively affect self-esteem and quality of life, leading to personality distortion and reduced self-confidence during childhood and adolescence. In severe cases where teenagers are not properly informed about how to seek help and recover from this pathological state, the negative impact can extend to psychological, physical, and social aspects of their lives, ultimately leading to poorer personal development and harming their affective and professional lives [30].

In the study, it was found that aerobic fitness was higher in the 13- and 16-year-old age groups, with a significant interaction between age and sex, with boys of all ages exhibiting greater aerobic fitness than girls. Non-fit rates were found to be higher in girls (42%) than boys (24%) for the 10- and 13-year-old age groups, with both groups having non-fit participants, which may negatively affect the development of heart problems and metabolic health that can persist into adulthood [31]. Low aerobic fitness during adolescence has been associated with abnormalities in cardiac function, reduced respiratory capacity, and decreased muscle strength [32]. Additionally, aerobic fitness has been associated with hormonal deregulation [33], autoimmune disease [28], childhood cancer [31], and increased risks of mental illness [32].

In a similar cross-sectional epidemiological study of 1223 Brazilian adolescents, Minatto et al. [34] found a proportion of non-fit girls of 51.3%, which was positively influenced by low economic status. Furthermore, a recent investigation revealed that poor aerobic fitness was associated with reduced adolescent neuroplasticity, cognitive function, and academic performance [35]. A healthy lifestyle, characterized by regular sports practice combined with a healthy diet, can help promote aerobic fitness during adolescence and prevent disease [30]. This evidence underscores the importance of consistently engaging in physical activity [9].

The research found that boys had greater abdominal resistance than girls, and that adolescents aged 13 and 16 of both sex had higher abdominal resistance than the 10-year-old group. Furthermore, 13- and 16-year-old boys exhibited greater abdominal strength than girls, except for 10-year-old girls. Abdominal resistance is an indicator of physical fitness, which has been linked to higher values. Conversely, aerobic resistance is a physical fitness indicator that is associated with metabolic syndrome, insulin resistance, and type 2 diabetes [36]. Sex differences in physical fitness emerge during puberty, and it is typical for boys to have a higher physical fitness index than girls due to the different moments of maturation [37]. However, the relevance of the sex difference in aerobic fitness during puberty is being questioned in view of the socioeconomic and environmental factors that influence it [11].

The findings of this study support this information because although boys had higher abdominal resistance on average, both groups had considerable numbers of participants classified as non-fit (boys: 27%; girls: 22%). Continuous sports practice can improve abdominal resistance, which can lead to core stability and better spinal positioning, resulting in a reduction in low back pain—an important factor for well-being and quality of life [38]. Additionally, maintaining core strength can help individuals to have better control over their body for everyday activities such as lifting and carrying, and can also facilitate participation in school sports [39].

Our study found that 16-year-old boys had greater upper limb resistance than girls. Interestingly, both sex had higher rates of unfit adolescents, with boys having a much higher rate of unfitness than girls. These findings are consistent with those of Fraser et al. [40], who conducted a 34-year longitudinal study of 8498 Australian schoolchildren aged 7 to 15 years and found that 26.9% of participants experienced declines in upper/lower limb and core strength over time. Upper limb resistance is associated with overall muscle strength and aerobic fitness, which are important determinants of quality of life [41].

In all the age groups, girls exhibited greater lower limb power than boys. Furthermore, adolescents aged 13 and 16 showed increased lower limb power regardless of sex. Non-fit girls in the 10-year-old age group had higher rates (24%) than boys (10%), represented by a moderate effect size. Conversely, in the 16-year-old age group, unfit boys had higher rates than girls, represented by a large effect size. Rauch et al. [42] analyzed 80 adolescents aged 8 to 18 years (42 girls and 38 boys) and found that obese adolescents had 24% lower limb force production compared to non-obese adolescents. Physical fitness is mainly influenced by behavioral factors, and regardless of differences in absolute scores, both sex can obtain health benefits through increases in muscle strength and power [43]. The muscular system is recognized as an endocrine organ that regulates metabolism, absorption, and energy expenditure, which are essential for preventing the accumulation of adipose tissue [44]. However, adolescents with decreased muscle functions are more likely to develop insulin resistance, diabetes, dyslipidemia, and heart disease [40]. Additionally, lower limb power is positively correlated with attention levels in preschoolers and has a positive effect on school performance [45].

In addition, our study found that 13- and 16-year-old boys and girls exhibited better running speeds, without any significant interaction between age and sex. Furthermore, boys aged 10 and 16 years showed higher rates of being physically unfit (with a small and moderate effect size, respectively) than girls, whereas girls aged 13 showed higher rates of being physically unfit (with a small effect size) than boys. A cohort study conducted by Vandoni et al. [46] on 3923 Italian adolescents aged 11–13 years found that overweight and obese adolescents exhibited a slower 5-meter repeated running speed compared to healthy participants. It is well-known that running speed is a physical ability that is directly associated with good body composition [47], better school performance [45], and protection against metabolic syndrome [48]. Therefore, the higher proportion of non-fit boys and girls (41% and 34%) in our study indicates overall health risks [26]. Additionally, maintaining the function of the lower limbs throughout life is crucial for preventing the progression of physical frailty in old age [49].

Moreover, in the bigger part of the results, boys presented better absolute physical fitness scores than girls. These results were expected, considering the well-described effects of pubertal maturation on typical sex discrepancies in physical fitness [23]. In contrast, the physical fitness cut-offs revealed that higher proportions of non-fit adolescents from both sex remain at risk for chronic non-communicable diseases [5]. Furthermore, the present study indicates that, possibly, both sex presented negative health behaviors at some point in their childhood and adolescence [8]. In addition, despite the moderate effect size influence favoring better physical fitness for boys, the prevalence of obesity in both sex were high (44% in boys and 41% in girls), without significant differences between sex rates, and higher than our previous findings (18% in boys, and 20% in girls) [13]. Regarding these inferences, it becomes clear that, beyond the statistical differences, the cut-offs reveal that there is still a great risk for the physical and mental health of adolescents [6].

### Limitations, Study Strengths, and Perspectives

The current study has several limitations that need to be acknowledged. Firstly, we did not assess the levels of physical activity, length of sedentary behavior, or habits of adolescents, which could provide valuable information about the relationship between these factors and obesity. Secondly, the sample size was relatively small, comprising only 85 men and 85 women, and it was limited to the northeastern region of Portugal. As a result, the study’s results may not be generalizable to the overall population as it only represents a specific region of Portugal. Furthermore, we did not investigate other potential influencing factors on adolescent obesity status and physical fitness levels. These include factors such as television viewing, parental obesity, birth weight, sleep duration, maternal education, being an only child, and family size, all of which can positively contribute to childhood obesity. Therefore, future studies should consider incorporating these relevant variables into their analysis models. Lastly, the study’s cross-sectional design limits the determination of causality. Cross-sectional studies only provide a snapshot of data over a short time interval, so they cannot establish temporal relationships between variables or determine cause-and-effect relationships. Future longitudinal studies are therefore required to confirm the current findings.

On the other hand, the current study has strengths that should be recognized. The multivariate analyses, including sex differences, in this dataset of Portuguese adolescents provide valuable information about the sex-specific aspects of obesity and physical fitness. This study serves as a valuable starting point for further analysis on more extensive and diverse datasets. Therefore, future studies with larger samples from different regions of Portugal and incorporating complementary measures, such as physical activity levels, time of sedentary behavior, nutritional status, family history, and socioeconomic factors, are recommended to obtain a better understanding of how these behavioral and environmental factors can modulate the state of obesity and physical fitness. Longitudinal designs can also contribute significantly to understanding the significance of changes in obesity status and physical fitness levels among Portuguese adolescents over time.

## 5. Conclusions

The present study has revealed important findings regarding the hypotheses tested. First, hypotheses (a) and (b) were contradicted, as the 13–16-year-old age groups displayed higher BMI scores and abdominal adiposity, irrespective of sex, and both boys and girls had high rates of obesity (44% in boys and 41% in girls). In regard to hypothesis (c), some aspects were confirmed, as the 13/16-year-old age groups displayed greater aerobic fitness regardless of sex. Boys exhibited better aerobic fitness than girls across all age groups and greater abdominal and lower limb resistance, as well as better upper limb resistance in the 16-year-old group. Additionally, boys displayed a better sprint ability than girls across all age groups. Lastly, although boys generally performed better in most measures of physical fitness, both sex had a high proportion of non-fit adolescents, with boys being the most affected.

## Figures and Tables

**Figure 1 ijerph-20-06115-f001:**
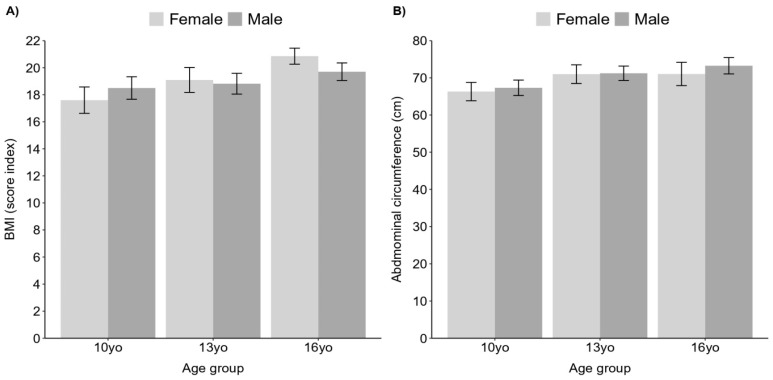
BMI (**A**) and abdominal circumference (**B**) across sex and age groups.

**Figure 2 ijerph-20-06115-f002:**
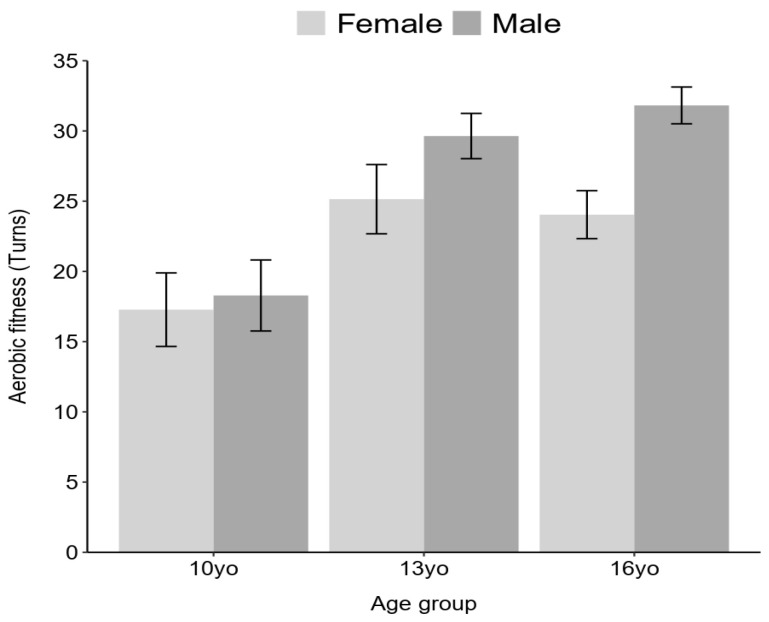
Aerobic fitness across sex and age groups.

**Figure 3 ijerph-20-06115-f003:**
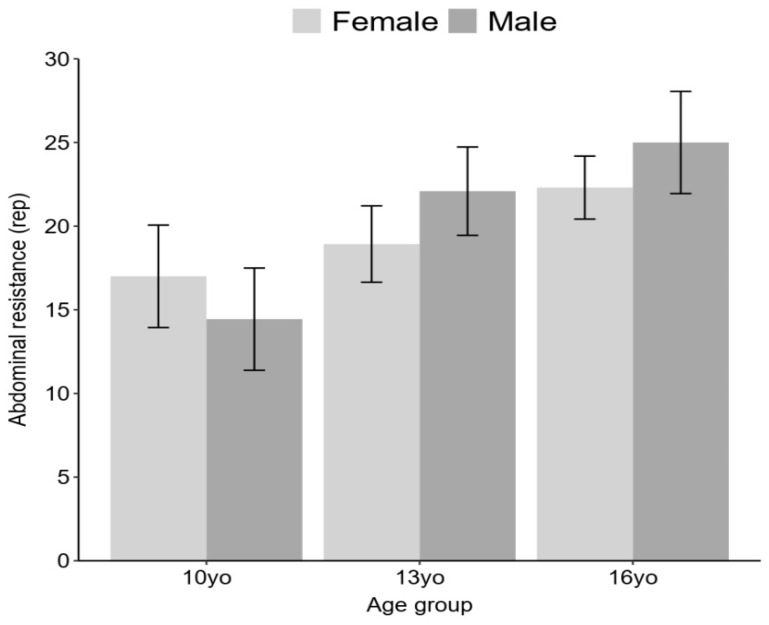
Abdominal resistance across sex and age group.

**Figure 4 ijerph-20-06115-f004:**
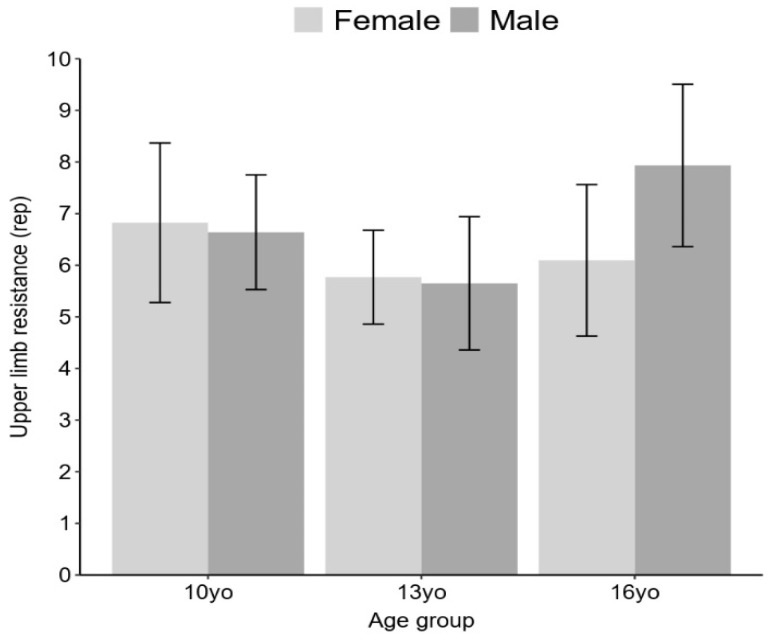
Upper limb resistance across sex and age.

**Figure 5 ijerph-20-06115-f005:**
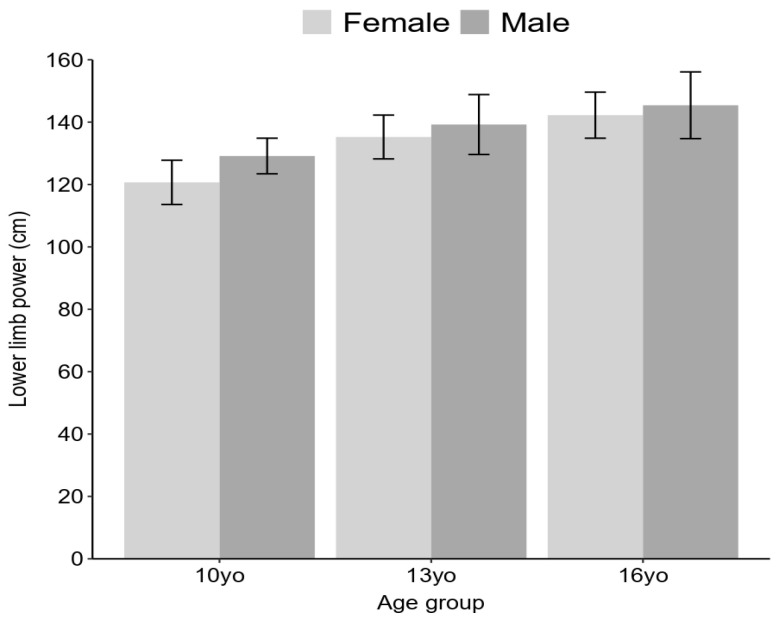
Lower limb power across sex and age groups.

**Figure 6 ijerph-20-06115-f006:**
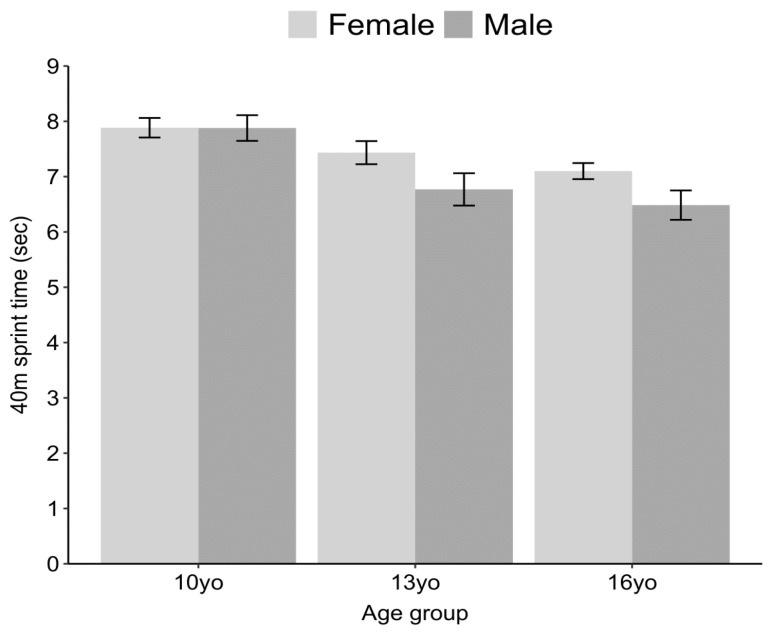
Maximum running speed across sex and age groups.

**Table 1 ijerph-20-06115-t001:** Sample characteristics.

Variables	Normal Weight (n)	Obesity (n)	X^2^	*p*-Value	ES
**Sex**					
**Male (n = 85)**	48 (56%)	37 (44%)	20.20	*p* < 0.001	0.15
**Girls (n = 85)**	50 (59%)	35 (41%)	31.40	*p* < 0.001	0.19
**BMI by age**					
**10 yo (n = 58)**	27 (46%)	31 (58%)	19.70	*p* < 0.001	0.22
**13 yo (n = 60)**	34 (57%)	26 (43%)	22.90	*p* < 0.001	0.23
**16 yo (n = 54)**	37 (71%)	15 (29%)	75.00	*p* < 0.001	0.48

Notes: n = sample size; X^2^ = chi-squared statistics; *p*-value = confidence level at 95% for statistical significance; ES = effect size.

## Data Availability

Data are available under request to the contact author.

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
