# Peer review of "Obesity Status and Physical Fitness Levels in Male and Female Portuguese Adolescents: A Two-Way Multivariate Analysis"

_ijerph, 2023, doi:10.3390/ijerph20126115_

Round 1

Reviewer 1 Report

General Comments

·         This study determined obesity and physical fitness levels of Portuguese adolescents through a multivariate analysis. It is interesting that the study documented significant gender and age main effect in most of the variables and a gender x age interaction effect in aerobic fitness. However, I do have some concerns, especially the results and discussion sections which are not properly presented.

·         There is also inadequacy of language command, particularly usage of correct grammar and spelling errors. This is the case throughout the manuscript.

·         Specific comments are presented below.

Specific Comments

Abstract

·         Lines 35 & 36: The sentence should read: association between gender and chronological age with obesity ….

·         Line 37: assessing, not accessing.

Introduction

·         The background information looks good, though verbose.

·         Page 2, Line 78: … socioeconomic may be one of the most …

Materials and Methods

2.1. Study participants

·         Page 3, Line 102: … association of power and age with obesity …

·         Provide information on sample size determination with a relevant formula.

·         Lines 117-123: After parental consent, minors only need to give assent and not consent again.

·         Page 3, Line 135: height was collected or measured?

·         Page 4, Lines 161-162: … the speed was slower, not lower

·         Page 4, Line 175-176: Confused sentence. Review the sentence please. Use participants instead of subjects.

·         Page 5, Line 192: Use restraining line instead of starting line.

·         Page 5, Line 209: … descriptive statistics were expressed as means …

·         Page 5, Line 231: The date 2013 is unnecessary. Delete it.

Results and Discussion

·         Pages 5 & 6: There is wrong use of statistical expressions. When there is a significant effect or difference, it should be written: P<0.05, and when there is no significant effect, it is P>0.05. Presentation should be properly done..

·         The gender x age significant interaction in aerobic fitness needs to be properly described.

·         Reporting of results is not logical and consistent.

·         Page 9, Line 363: Are BMI, abdominal circumference, aerobic fitness, and others independent variables?

·         The discussion is not well articulated, no comparison with previous studies. No information on whether findings were supported or not supported with previous studies.

·         One of the limitations of this study is the cross-sectional design which precluded determination of causality. This should be included.

Author Response

We appreciate your positive feedback. Point-by-point responses as provided, and revisions are tracked in the manuscript using the track change option in MS Word.

Reviewer 2 Report

Dear,

please find the comments attached.

Sincerely,

reviewer

Author Response

(The authors gave the same response as above.)

Round 2

Reviewer 1 Report

The Discussion section needs to be improved upon.

Author Response

Review: The Discussion section needs to be improved upon.

Response 1: Dear reviewer. We very much appreciate your suggestions. Following your suggestions, we improved the discussion.
